# The fractal turbulent/non-turbulent interface in the atmosphere

Lars Neuhaus, Matthias Wächter, and Joachim Peinke

ForWind and Institute of Physics, School of Mathematics and Science, Carl von Ossietzky Universität Oldenburg, Ammerländer Heerstraße 114–118, 26129 Oldenburg, Germany

**Correspondence:** lars.neuhaus@uol.de

**Abstract.** With their constant increase in size, wind turbines are reaching unprecedented heights. Therefore, at these heights, they are influenced by wind conditions that have not yet been studied in detail. With increasing height, a transition to laminar conditions becomes more and more likely. In this paper, the presence of the turbulent/non-turbulent interface (TNTI) in the atmosphere is investigated. Three different on- and offshore locations are investigated. Our fractal scaling analysis leads to typical values known from ideal laboratory and numerical studies. The height distribution of the probability of the TNTI is determined and shows a frequent occurrence at the height of the rotor of future multi megawatt turbines. The indicated universality of the fractality of the TNTI allows the use of simplified models in laboratory and numerical investigations.

## 1 Introduction

Wind turbines are getting bigger and bigger, reaching heights of over $250\,\mathrm{m}$, and are installed farther offshore. The turbulent wind at these locations and heights is rarely measured. Therefore, the environmental conditions for future offshore wind turbines are still poorly understood. However, these conditions have a significant impact on the performance of wind turbines. It is known that wind fluctuations on short time scales cause fluctuations in the power output of wind turbines (Milan et al., 2013). In addition, a varying turbulence intensity (TI) of the inflow over the rotor also has a significant influence on turbine operation (Lobo et al., 2023).

With the new developments in wind energy, the transitions from turbulent to laminar conditions are becoming increasingly important. In particular, the complexity of these turbulent/non-turbulent interfaces (TNTI) can have an impact on working conditions, which is the focus of our paper.

While the TNTI has been extensively studied in laboratory flows, it has hardly been investigated in the atmosphere. Available data covers heights up to $100\,\mathrm{m}$ offshore using met masts such as the FINO platforms and up to $200\,\mathrm{m}$ onshore using met masts such as the Cabauw met mast. More Extreme heights up to $250\,\mathrm{m}$ offshore are measured using Lidar systems, which however provide lower temporal resolution. Recently, flights have been carried out to investigate the turbulence around wind parks, covering different heights (Lampert et al., 2020). However, flights only allow a short observation period and can only provide limited picture regarding turbulent properties.

The question arises whether we can find similarities between the characteristics of TNTI from ideal laboratory and numerical studies and those from atmospheric situations. The objective of this paper is to make a first characterization of atmospheric data in order to identify the TNTI in the atmosphere and compare it on the basis of known features, namely fractal characteristics.

The aim is therefore not to discuss minor details, but to provide a basic idea of the presence of the TNTI in the atmosphere and the possibilities of characterizing it.

The applied method is described in detail for measurements at the FINO1 platform and additional sites are investigated to provide a more complete picture. The measurement sites considered are described in Sect. 2. The basic features and methods of characterizing the TNTI are presented in Sect. 3. The results of the analysis are presented in Sect. 4 and discussed in Sect. 5. Sect. 6 concludes this paper.

## 2 Measurement sites

Three different sites with height resolved data are used for the analysis. The FINO1 met mast, the Cabauw met mast (Lidar measurments available), and Lidar measurements at the offshore platform Borssele Alpha are taken into account.

The FINO1 offshore met mast has a height of 103m (FINO1, 2023). It is selected for a detailed discussion, as it is a well known offshore platform, which provides temporal high resolved data on a long observation period. Cup anemometer at $33\,\mathrm{m}$, $40\,\mathrm{m}$, $50\,\mathrm{m}$, $60\,\mathrm{m}$, $70\,\mathrm{m}$, $80\,\mathrm{m}$, $90\,\mathrm{m}$, and $100\,\mathrm{m}$ record the wind speed simultaneously with a sampling frequency of 1Hz. Wind vanes at 33m and 90m record the wind direction. As for certain inflow directions the mast influences the measurements, data for wind directions between $275°$ and $350°$ of either directional sensor are neglected (filled with NaNs) to ensure undisturbed inflow. Further, as FINO1 is located next to several wind parks, only data up to the date of the assembly of the first wind turbine are considered. The available time period is hence from 01.01.2007 to 15.07.2009. Further, low wind speeds ($u < 0.5\,\mathrm{m\,s^{-1}}$), which tend to unreasonable high TI and which have minor importance for the operation of wind turbines, are neglected (data filled with NaNs).

The Cabauw wind data were made available by the Royal Netherlands Meteological Institute (KNMI) (Hansen et al., 2021). The $213\,\mathrm{m}$ high met mast is installed onshore. Propeller anemometer at $20\,\mathrm{m}$, $40\,\mathrm{m}$, $80\,\mathrm{m}$, $140\,\mathrm{m}$, and $200\,\mathrm{m}$ record the wind speed simultaneously with a sampling frequency of 2Hz. For the time period from 1985 to 1986 roughly 480 hours are available.

At the same site Lidar measurements were conducted and made available by KNMI (KNMI, 2023a, b). Two data sets recorded by a ZephIR 300M wind lidar are avaibale. One data set (Cabauw Lidar ZP) includes wind speeds at seven heights from $10\,\mathrm{m}$ to $251\,\mathrm{m}$ (not equidistant) with a temporal resolution of about $11\,\mathrm{s}$. This data is available in the time period from 15.02.2018 to 07.06.2020. The second data set (Cabauw Lidar ZX) includes wind speeds at eleven heights from $10\,\mathrm{m}$ to $299\,\mathrm{m}$ (not equidistant) with a temporal resolution between $17\,\mathrm{s}$ and $18\,\mathrm{s}$. This data is available in the time period from 20.02.2020 to 07.06.2020.

Further, data by Lidar measurements at the Borssele Alpha offshore platform (BSA) next to the wind park Borssele I-V (operation started in September 2021) is used which was also made available by KNMI (KNMI, 2023c). Data was recorded by a ZephIR 300M wind lidar at 11 heights from $14\,\mathrm{m}$ to $249\,\mathrm{m}$ (not equidistant) with a temporal resolution between $17\,\mathrm{s}$ and $18\,\mathrm{s}$. Data is available from end of 2019 until now. Measurements are still ongoing. The considered time period in this paper is from 21.11.2019 to 31.08.2021.

## 3 Method

In this section, the approach used in this work to detect and to characterize the TNTI is presented. A brief introduction to laboratory experiments on the TNTI is given (Sect. 3.1). Section 3.2 describes the characterization of boundaries based on the fractal dimension. In Sect. 3.3, a method for characterizing the TNTI in the atmosphere based on one-point measurements of the wind velocity is given and shown exemplarily for the FINO1 site.

### 3.1 Turbulent/non-turbulent interface (TNTI)

Between different flow states, as turbulent and non-turbulent, an interface forms. An example of a turbulent/non-turbulent interface (TNTI) of a jet is shown in Fig. 1. The mixing of the two flow phases occurs on large and small scales. It can be recognized how the complexity of this interface increases downstream.

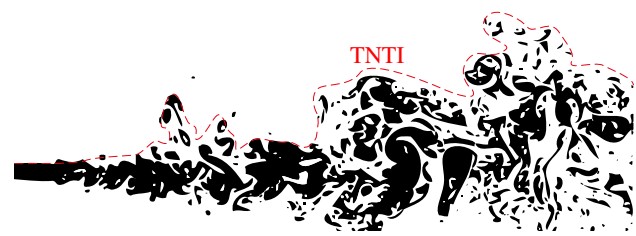

**Figure 1.** Jet flow visualized by laser induced fluorescence (adapted from Dimotakis et al. (1981); Sreenivasan and Meneveau (1986)). The fractal boundary between the turbulent flow and the laminar surrounding can be seen. It is here indicated by a red line, which is an approximation of the TNTI.

The TNTI was first investigated in laboratory flows by Corrsin and Kistler (1955). Sreenivasan and Meneveau (1986) were the first to describe the boundary between laminar and turbulent flows through its fractal dimension. They investigated a 70 developing turbulent boundary layer on a flat plate. The turbulent flow was made visible by smoke. Images were taken and a brightness threshold was used to determine the TNTI. By changing the image resolution, different scales were resolved and on an intermediate range of scales between the Kolmogorov length scale and $1/6$ integral length scale $L$ a fractal dimension of the TNTI of about 2.4 was found.

Following this work, more detailed studies were carried out using more sophisticated methods such as Particle Image Ve-75 locimetry (PIV). de Silva et al. (2013) used PIV measurements to detect the TNTI in a boundary layer flow using a turbulent kinetic energy (TKE) threshold. They found a more precise fractal dimension of 2.36 on scales from $20\%$ $L$ to the smallest scales (limited by the resolution). Based on these results, we define a fractal dimension of a TNTI of 2.36 (or 0.36 for a one-dimensional cut, as explained below) as a typical TNTI fractal dimension of a TNTI.

While the TNTI itself is rather thin, its position is strongly varying. The TNTI is formed on large scales by engulfment (large 80 scale fluctuations of the interface) and on small scales by nibbling (viscous diffusion process). From large to small scales, the

TNTI exhibits a self similarity that can be found in the fractal dimension. A detailed review on the TNTI is given by da Silva et al. (2014) and a more recent summary is given by Xu et al. (2023).

## 3.2 Fractal dimension

This turbulent/non-turbulent interface is commonly described by its fractal characteristics. Fractals were intensively studied by Mandelbrot (1982) and became an object of interest for the scientific community. To characterize a fractal its fractal dimension can be used.

An exemplary fractal curve that corresponds to a boundary in two-dimensional space is the Koch curve (Fig. 2). The construction scheme consists of replacing the middle subinterval of an interval with two equally sized subintervals. From the resulting intervals, the middle subintervals are again replaced by two subintervals of the same size, and so on to smaller and smaller intervals (increasing order $n$). The result is a fractal boundary, which in this case follows a strict geometric law.

The fractality of this Koch curve can be estimated by a box counting approach, which results in the fractal dimension (box-counting dimension or Minkowski–Bouligand dimension). To do so, boxes with different edge length $r$ are used and the number of boxes $N(r)$ required to cover the curve is counted. The fractal dimension $D_f$ (box-counting dimension) can then be determined by the slope of the relation

$$N(r) \propto r^{-D_f} \tag{1}$$

to 1.262 for the Koch curve (see Sreenivasan and Meneveau, 1986).

In real-world applications, data with a high spatial resolution is not always available. Atmospheric data in particular is mostly only available by point wise measurements. The limited amount of vertical measurement points of the investigated data sets is not sufficient for a two-dimensional analysis. However, by Taylor's hypothesis of frozen turbulence (Taylor, 1938) the individual point measurements will give a one-dimensional slice through a three-dimensional field. By the additive rule of

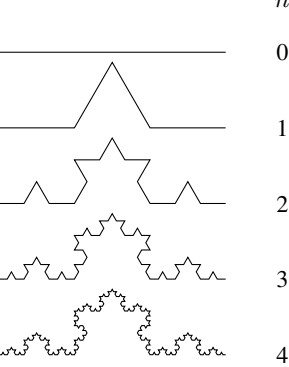

**Figure 2.** Koch curve of the order $n$.

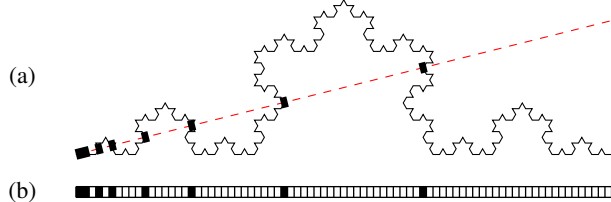

**Figure 3.** Koch curve f order 4 with one-dimensional slice and marked intervals of crossings with the Koch curve (a). The corresponding intervals give the Cantor set like plot (b).

co-dimensions for intersecting sets

$$D_{f,3} = D_{f,2} + 1 = D_{f,1} + 2 \tag{2}$$

the fractal dimension $D_{f,d}$ in higher embedding dimensions $d$ can be estimated by data collected in a lower embedding dimensions (see Mandelbrot, 1982; Sreenivasan and Meneveau, 1986). Furthermore, the fractal dimension is bounded by the embedding dimension $d$ and the corresponding lower dimension $d-1$, e.g. a smooth surface in three-dimensional space would scale with $D_f = 2$, whereas a space-filling surface would exhibit a fractal dimension of $D_f = 3$.

Thereby, a simple way to estimate the fractal characteristic of a boundary in three-dimensional or two-dimensional space is to consider a one-dimensional slice (e.g. a single point measurement of the wind speed). This slice (red dashed line in Fig. 3 (a)) is covered with intervals of size $r$ and intervals with and without a boundary crossing are obtained (as indicated in Fig. 3 (b)). The fractal dimension of this slice can be estimated after Eq. 1 by the number of intervals $N(r)$ on the scale $r$ that are needed to cover the boundary crossings. The result of this box counting approach is $D_{f,1} = 0.262$ and after Eq. 2 gives the correct $D_{f,2} = 1.262$.

This clearly shows that the fractal dimension of higher-dimensional fractals can be estimated from a one-dimensional slice. Consequently, an adequate estimate of the fractality of the TNTI in the atmosphere can be made from the available single point measurements, which correspond to a slice through a three-dimensional wind field.

### 3.3 Applied method

Typically, when applying methods to calculate the fractal dimension, the challenge lies in determining the interface using a threshold. Details on the herein applied method are discussed for the FINO1 site. A similar procedure as in de Silva et al. (2013) is used to estimate the TNTI. To determine the TNTI, the instantaneous turbulent kinetic energy (TKE) is used to detect transitions between laminar and turbulent phase. Subsequently, the box counting approach just mentioned is applied to characterize the TNTI by its fractality.

The instantaneous TKE is approximated by

$$E = \frac{1}{2}(u - u_{\text{movavg}})^2 \tag{3}$$

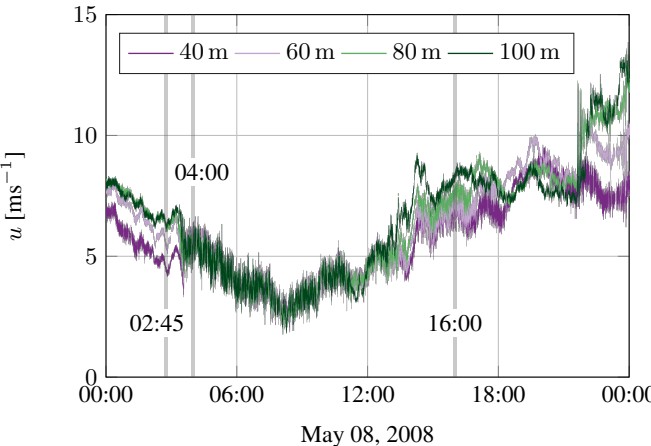

**Figure 4.** Exemplary velocity time series at FINO1 on May 08, 2008 for different heights. Vertical gray bars indicate exemplary 10 minute sections shown in Fig. 5 and Fig. 6.

with the moving averaged wind speed

$$u_{\mathrm{movavg}} = \frac{1}{Tf_s} \sum_{\Delta t=-T/2}^{T/2} u(t+\Delta t). \tag{4}$$

Here the sampling frequency $f_s$ and the filter span $T$ of $20\,\mathrm{s}$ (for cup and propeller anemometer) and $90\,\mathrm{s}$ (for Lidar measurements) are used. These values are chosen, as they mark the boundary between 3D turbulence and large scale fluctuations (see Sim et al., 2023). For the lidar measurements a larger window size is considered as a compromise between a sufficient amount of samples for the estimation of the TKE and sufficiently small scales. To validate that choice, we performed a study on the influence of a variation of $T$, which showed no significant changes for $T > 20\,\mathrm{s}$ and thus shows a robust behavior for changes on large scales (see Appx. A).

For better comparison of different mean wind speeds, the instantaneous TKE is normalized

$$E_{\mathrm{norm}} = E/u_{\mathrm{movavg}}^2 \tag{5}$$

by the square of the moving averaged wind speed. The threshold between turbulent and non-turbulent phase is set to $0.001$, which is in the order of the threshold used by de Silva et al. (2013). Data points where this threshold is crossed will be referred to as crossings in the following.

The next steps are shown examplarily for a day (May 8, 2008) of the FINO1 data set (Fig. 4), as this day exhibits many laminar periods. The investigation is done for sections of 10 minute length (sensitivity on section length is shown in Appx. B). In Fig. 5 crossings of the TNTI are visualized for the different heights.

Figure 5 (a) shows the behavior of a rather turbulent 10 minute section. Plenty crossings can be observed at different heights. This is not always the case as shown by the selected section of Fig. 5 (b) and (c). In Fig. 5 (b) a laminar phase at high altitudes

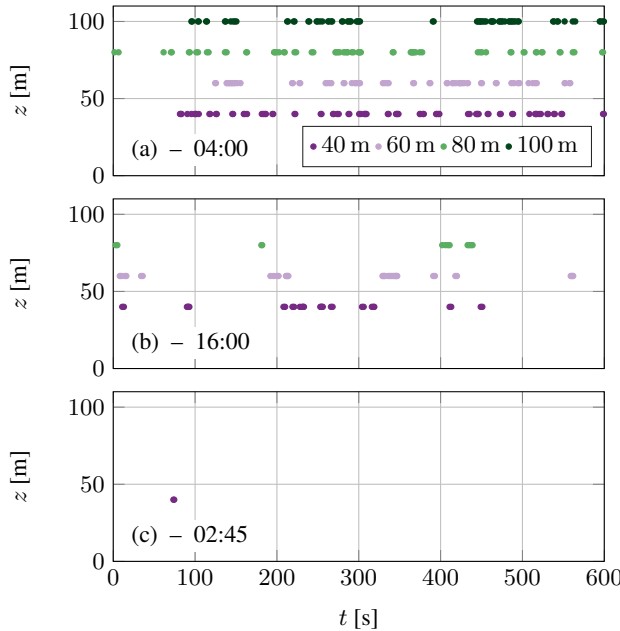

**Figure 5.** Crossings indicating the transition between laminar and turbulent phase for an exemplary turbulent (May 08, 2008 04:00) (a), turbulent/non-turbulent (May 08, 2008 16:00) (b) and laminar (May 08, 2008 02:45) (c) section. A Cantor set like plot as shown in Fig.3 (b).

(100 m) with no crossings is shown whereas at lower altitudes crossings can be recognized. Figure 5 (c) shows the behavior of a section with laminar flow at all altitudes. Almost no crossings of the threshold occur.

To estimate the fractal dimension (Eq. 1), our box counting approach is applied for each individual 10 minute section
for each height. Boxes of a certain size $r_{\text{Box}}$ (respectively duration $T_{\text{Box}}$) are used. Taylor's assumption of frozen turbulence $r_{\text{Box}} = \langle u \rangle T_{\text{Box}}$ is used to convert the time dependence into a spatial scale dependence (Taylor, 1938).

Next, the number of boxes with at least one crossing of the threshold is counted. After Eq. 1 the resulting number of counted boxes $N_{\text{Box}}$ over box size $r_{\text{Box}}$ is plotted in a double logarithmic presentation (Fig. 6). To improve the quality of the estimated slope, the boxes overlap by 90%.

It can be recognized, that mainly three different slopes can be found. A slope of $-1$ is found for fully turbulent behavior as shown in Fig. 6 (a). A slope of $-0.36$ was found for sections with turbulent and laminar phases (Fig. 6 (b)). For sections with mostly laminar flow, the slope is close to 0 (Fig. 6 (c)). Note that the scaling ranges for different exponents do not always extend over the entire range, but are often only limited to some sub-ranges of the scales, as can be seen in Fig. 6 (a,b).

The fractal dimension is determined by the negative slope of the just discussed presentation. The selection of the sub-range
of scales is motivated by our wind energy application. We take scales from roughly $3\,\text{m}$ to $250\,\text{m}$ corresponding to the order of a wind turbine chord length and rotor diameter, respectively.

Not for all 10 minute sections a clear slope is obtained. Sometimes there is a superposition of different slopes. For our purpose here, we consider such events as ranges without self-similarity (constant slope). To do so, 10 minute sections that

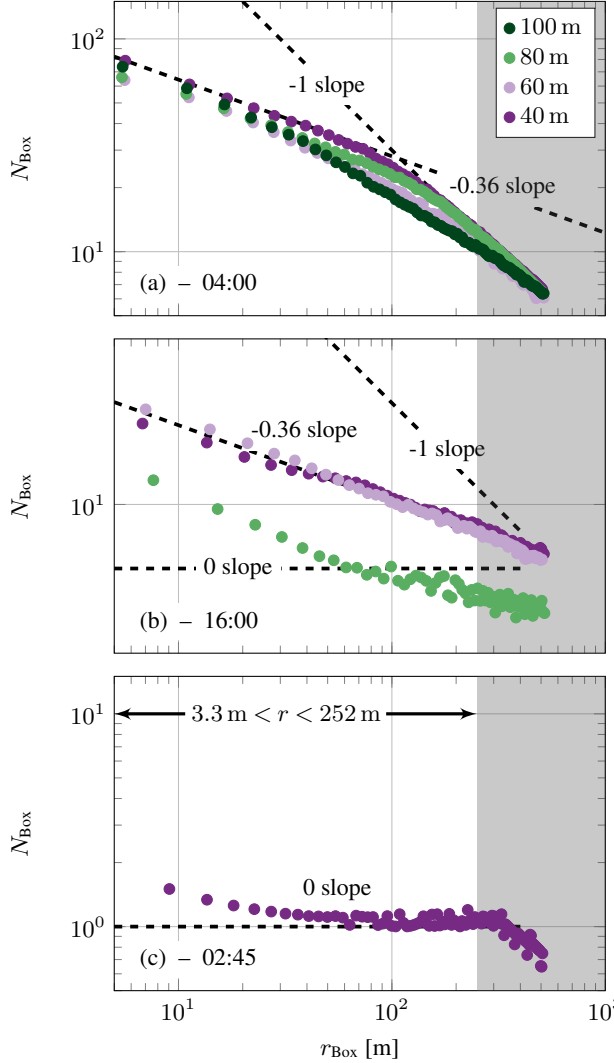

**Figure 6.** Number of boxes containing at least one threshold crossings $N_{\mathrm{Box}}$ as a function of the box size for three 10 minute sections around May 08, 2008 04:00 (turbulent) (a), May 08, 2008 16:00 (TNTI) (b), and May 08, 2008 02:45 (laminar) (c), according to Fig. 5.

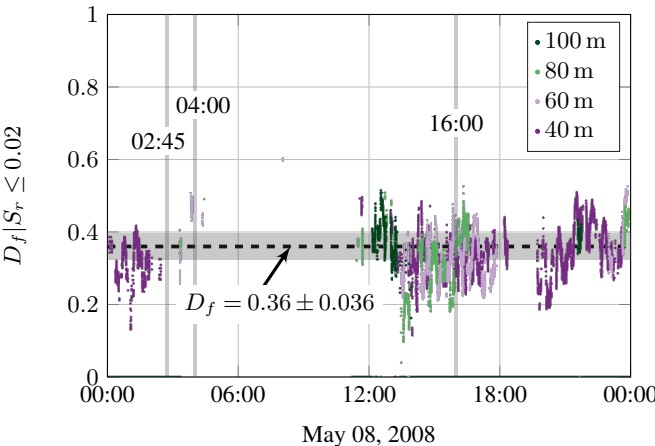

**Figure 7.** Estimated fractal dimension $D_f$ on May 08, 2008. Only results are shown when a reasonable fractal dimensions could be determined. Different colors stand for different heights. The dashed line indicates the typical TNTI fractal dimension of 0.36 and the shaded gray area a range of $\pm 0.036$ around this value. Vertical gray bars indicate exemplary 10 minutes section shown in Fig. 5 and Fig. 6.

**Table 1.** Fractal dimension $D_f$ and residual standard error $S_r$ on May 08, 2008 at different times and for different heights. Valid entries ($S_r \leq 0.02$) are shown in a **bold** font and neglected entries ($S_r > 0.02$ or NaN) are shown in an *italic* font.

| Time | $z$ [m] | $D_f$ | $S_r$ |
|---|---|---|---|
| 04:00 – Fig. 5 (a) & Fig. 6 (a) | *100* | *0.5246* | *0.0328* |
| | *80* | *0.4297* | *0.0407* |
| | *60* | *0.4774* | *0.0409* |
| | *40* | *0.4653* | *0.0460* |
| 16:00 – Fig. 5 (b) & Fig. 6 (b) | *100* | *NaN* | *NaN* |
| | *80* | *0.3912* | *0.0350* |
| | **60** | **0.3780** | **0.0096** |
| | **40** | **0.3063** | **0.0127** |
| 02:45 – Fig. 5 (c) & Fig. 6 (c) | *100* | *NaN* | *NaN* |
| | *80* | *NaN* | *NaN* |
| | *60* | *NaN* | *NaN* |
| | *40* | *NaN* | *NaN* |

have residual standard errors $S_r$ of the slope greater than 0.02 are neglected (NaN). By this only sections with a constant
fractality over roughly two decades are considered. For the exemplary day (May 08, 2008) the resulting time series of the fractal dimension $D_f$ are shown in Fig. 7. The values for the three exemplary times (04:00, 16:00, and 02:45) are given in Table 1.

## 4 Results

The analysis of the measurement sites is done in three steps. First a basic analysis of the turbulence intensity at the different sites and heights is done (Sect. 4.1). In the following the presence of a typical TNTI fractal dimension is investigated (Sect. 4.2). Last, the likelihood of the presence of the TNTI and its fractal dimension at certain heights is investigated for all sites (Sect. 4.3).

### 4.1 Turbulence intensity

The turbulence intensity

$$\text{TI} = \sigma(u_{\text{detrend}})/\langle u \rangle \tag{6}$$

is calculated by the standard deviation $\sigma$. $u_{\text{detrend}}$ denotes the velocity timeseries detrended by a linear fit, and $\langle u \rangle$ denotes the mean wind speed for a section of 10 minute length. Only 10 minute sections with at least 75% valid data are considered.

Figure 8 shows the resulting probability density functions (PDF) for the individual sites. All sites show an increase of low turbulence intensity sections with height.

The general trend towards a lower TI at higher altitudes is illustrated by a decrease of the median TI (med(TI)) and an increase in the portion of 10 minute sections with TI$< 1.5\%$ as a function of $z$ (Fig. 9, see Appx. C for an analysis of the intermittency factor $\gamma$). The measurements at FINO1 revealed the lowest median TI. Compared to the two offshore sites (FINO1 and Borssele), the measurements at the onshore site Cabauw show a significant higher TI at lower altitudes. The lidar measurements (Cabauw Lidar ZP, Cabauw Lidar ZX, and Borssele) show comparable curves. However, a direct comparison is difficult due to the different measurement methods, the different measurement periods and seasons. Thus these statistics are based on different meteorological conditions which were selected.

### 4.2 Fractal dimension of the TNTI

Next the fractal dimension of the TNTI is investigated for 10 minute sections with an overlap of $9\,\text{min}$. Figure 10 shows the individual probability density function (PDF) of the fractal dimension $D_f$ for different TI ranges. The PDFs are normalized including invalid fractal dimensions ($S_r > 0.02$), which are not shown but would correspond to a peak in the PDF at "NaN".

As shown in Fig. 10 (a), for a low TI ($< 2.5\%$), most found fractal dimensions are smaller than the expected typical TNTI fractal dimension of 0.36 (see Sect. 3.1). This is in accordance with Fig. 6, as laminar phases tend to exhibit a slope closer to 0.

For medium TI ($2.5\% < \text{TI} < 7.5\%$), significantly more valid fractal dimensions are found. As seen in Fig. 10 (b), the found values match well with the expected value of 0.36. Further, a clear height dependence can be found with more 10 minute sections with a typical TNTI fractal dimensions at higher altitudes.

For high TI ($< 7.5\%$) only few valid fractal dimensions are found, see Fig. 10 (c). One peak in the PDF can be recognized at values slightly above the typical TNTI fractal dimension and one even smaller peak close to 1. Again in good agreement with Fig. 6, as turbulent sections tend to exhibit slopes closer to 1.

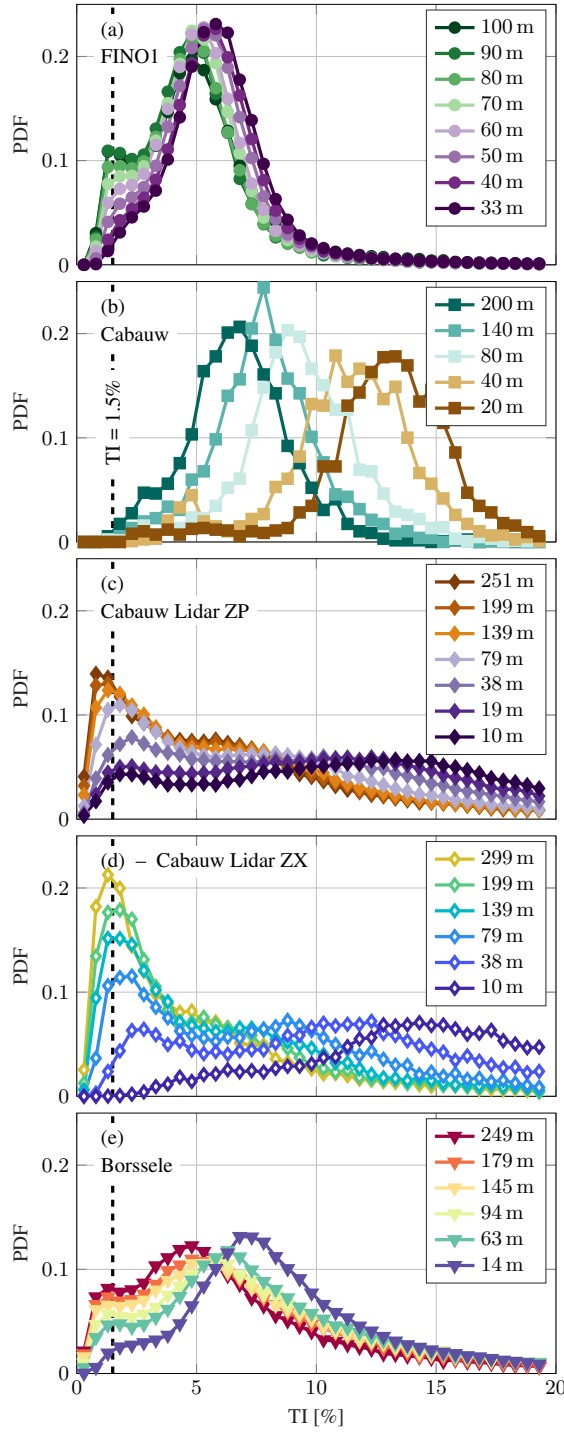

**Figure 8.** Probability density functions (PDF) of the turbulence intensity at different heights for the data sets FINO1 (a), Cabauw (b), Cabauw Lidar ZP (c), Cabauw Lidar ZX (d), and Borssele (e). For a further quantification, see Fig. 9.

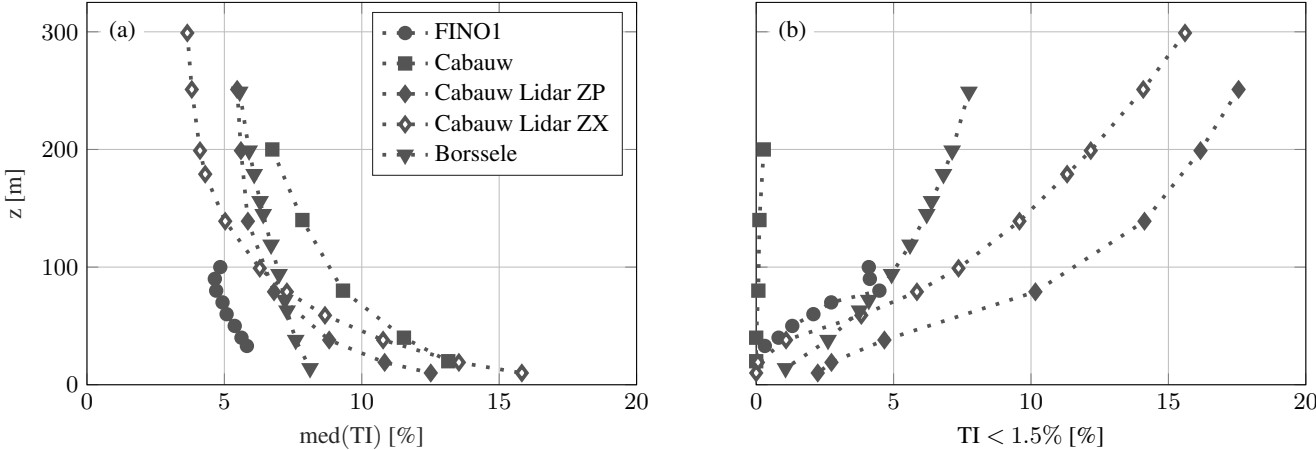

**Figure 9.** Median of the turbulence (a) and proportion of 10 min section with TI$< 1.5\%$ (b) at different heights and for the different data sets.

In Fig. 11 different probabilities of sections with a fractal dimension $D_f = 0.36$ within a $\pm 10\%$ range are shown. These probabilities are conditioned on the 10 minute section TI (Fig. 11 (a)), the mean wind speed $\langle u \rangle$ (Fig. 11 (b)), and the shear exponent $\alpha$ (defined later) (Fig. 11 (c)).

For periods with low TI ($< 2.5\%$) and high TI ($> 7.5\%$) only few events with a typical TNTI fractal dimension can be recognized (Fig. 11 (a)). For sections with TIs in between ($2.5\% < \text{TI} < 7.5\%$), it is more likely to exhibit both (laminar and turbulent) phases. Up to 17% of these observed 10 minute sections showed a typical TNTI fractal dimension.

At low mean wind speeds the percentage of sections with typical TNTI fractal dimension is rather indifferent over height (Fig. 11 (b)). This changes with increasing mean wind speed. A typical TNTI fractal dimension becomes more likely at higher altitudes and less likely at lower altitudes. However, for high mean wind speeds ($> 15\,\mathrm{m\,s^{-1}}$) the probability for a typical TNTI fractal dimension is reduced at all heights.

Figure 11 (c) shows results from data set conditioned on the shear exponent $\alpha$. $\alpha$ is estimated for all 10 minute sections by fitting the power law formulation $u(z) = u(z_\mathrm{ref})\left(\frac{z}{z_\mathrm{ref}}\right)^\alpha$ were $z_\mathrm{ref}$ is given by the highest altitude. Again, the probability of a typical TNTI fractal dimension becomes more likely with height. With increasing shear the probability of a typical TNTI fractal dimension has a maximum at altitudes around $60\,\mathrm{m}$ and decreases at higher altitudes. For extreme shear ($\alpha > 0.3$), the likelihood of a typical TNTI fractal dimension at higher altitudes ($90\,\mathrm{m}$) is reduced by half compared to lower shear ($\alpha < 0.3$).

Overall these probability investigations show that the occurrence of typical TNTI fractal dimensions are not negligible, but often are higher then 10% of the data.

## 4.3 Universality

Next an overview of results from all data sets is given. For the lidar measurements the estimation of the fractal dimension is adapted due to the lower sampling rate. The 10 minute sections is extended to 90 minute sections and the fractal dimension is

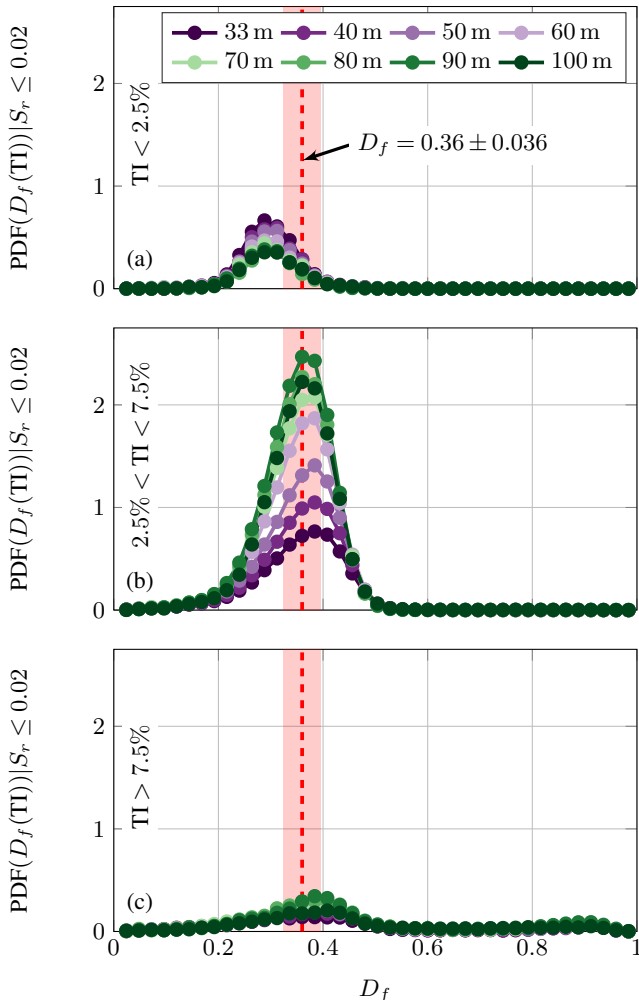

**Figure 10.** Probability density function of the fractal dimension $D_f$ conditioned on the different TI ranges: TI $< 2.5\%$ (a), $2.5\% <$ TI $<$ $7.5\%$ (b), and TI $> 7.5\%$ (c). The red dashed line indicates the typical TNTI fractal dimension of 0.36 and the shaded red area a range of $\pm 0.036$ around this value. The normalization of the PDFs is done based on all sections including invalid fractal dimensions ($S_r > 0.02$), which are not shown but would correspond to a peak at "NaN". For a further quantification, see Fig. 11 (a).

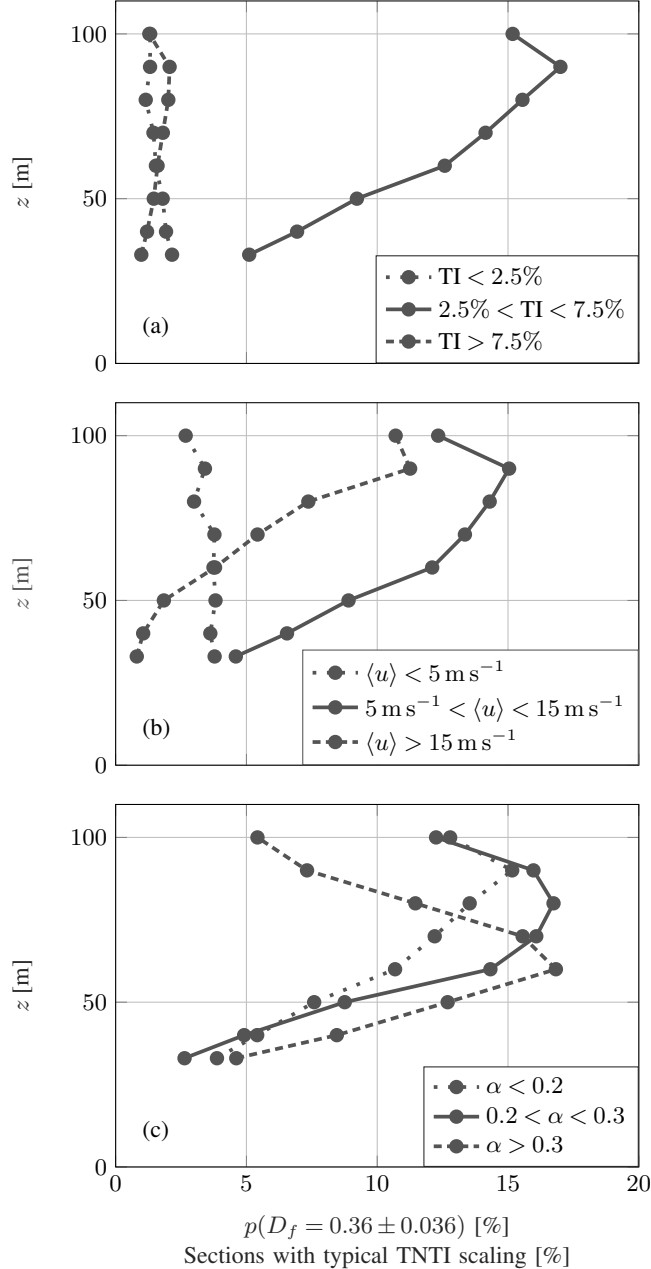

**Figure 11.** Percentage of data exhibiting a typical TNTI fractal dimension conditioned on different TI level (a), mean wind speeds (b), and shear (c).

estimated for scales from $200\,\mathrm{m}$ to $2.5\,\mathrm{km}$. Thus, the lidar measurements are used to investigate the presence of the TNTI on larger scales.

Figure 12 shows the distribution of the fractal dimension for each data set, according to Fig. 10 (b) for $2.5\% < \mathrm{TI} < 7.5\%$. The PDFs are normalized including invalid fractal dimensions ($S_r > 0.02$), which are not shown but would correspond to a peak in the PDF at "NaN". An accumulation of the fractal dimension can be observed for all data sets. However, some deviations can be recognized. At lower heights, a stronger deviation towards larger or smaller fractal dimensions can be recognized in the lidar measurements (Fig. 12 (c-e)). For more extreme heights, the fractal dimension tends to be closer to the typical TNTI fractal

dimension of 0.36. However, a broader distribution and shifts towards higher and lower fractal dimensions can be observed.

     The propeller measurements at Cabauw show only few events with a slightly towards lower values shifted fractal dimensions (Fig. 12 (b)). The results at $20\,\mathrm{m}$ are questionable and might be effected by ground structures.

     In contrast to the other data sets, the best values for the Cabauw Lidar ZP are obtained for $10\,\mathrm{m}$ with $0.2 \pm 0.1$ (Fig. 12 (c)). The peak of the fractal dimension gets more smeared out as the heights increase.

The results from Cabauw Lidar ZX show a consistent trend from which only the low altitude deviates (Fig. 12 (d)). With increasing height the peak of the fractal dimension becomes narrower and is shifted towards lower fractal dimensions from 0.56 at $38\,\mathrm{m}$ to 0.43 at $299\,\mathrm{m}$.

     Also the results from Borssele show a consistent picture with clearer and more frequent fractal structures at higher altitudes (Fig. 12 (e)). However, the fractal dimension peak is at 0.46 and hence higher than the expected typical TNTI fractal dimension

of 0.36.

     For all sites and data sets it can be recognized, that the probability of the typical TNTI fractal dimension ($0.324 \geq D_f \leq 0.396$) increases with height (Fig. 13). The obtained probabilities depend on sites and measurement methods. The FINO1 data set shows the highest ratio of typical TNTI fractal dimension. For the Cabauw site the dependence on different measurement methods or, respectively, time resolution of the measurements, is seen.

## 5    Discussion

A frequent presence of the turbulent/non-turbulent interface (TNTI) in the atmospheric data is observed. The presented method provides information on how frequently TNTI features occur at the investigated heights, but does not allow the height position of the TNTI to be determined. For the most reliable data set FINO1 with a high temporal resolution and a long observation period, a clear accumulation of the fractal dimension of this TNTI around 0.36 is found. To our interpretation this is in good

agreement with experiments in the laboratory (see de Silva et al., 2013).

     If investigating the individual sections of a data set, fractality (self similarity) on different scales can be observed. The box counting approach showed mainly three different slopes, 1 for fully turbulent flow, 0.36 for the TNTI, and 0 for fully laminar flow (Fig. 6). The slopes are not necessary constant on different scales. Different slopes on different scale ranges can be present (see Sreenivasan and Meneveau, 1986). When conditioning on the fit quality by the residual standard error, mostly the typical

TNTI fractal dimension of 0.36 is observed (Fig. 10). By this approach only fractal dimensions with a constant fractality over

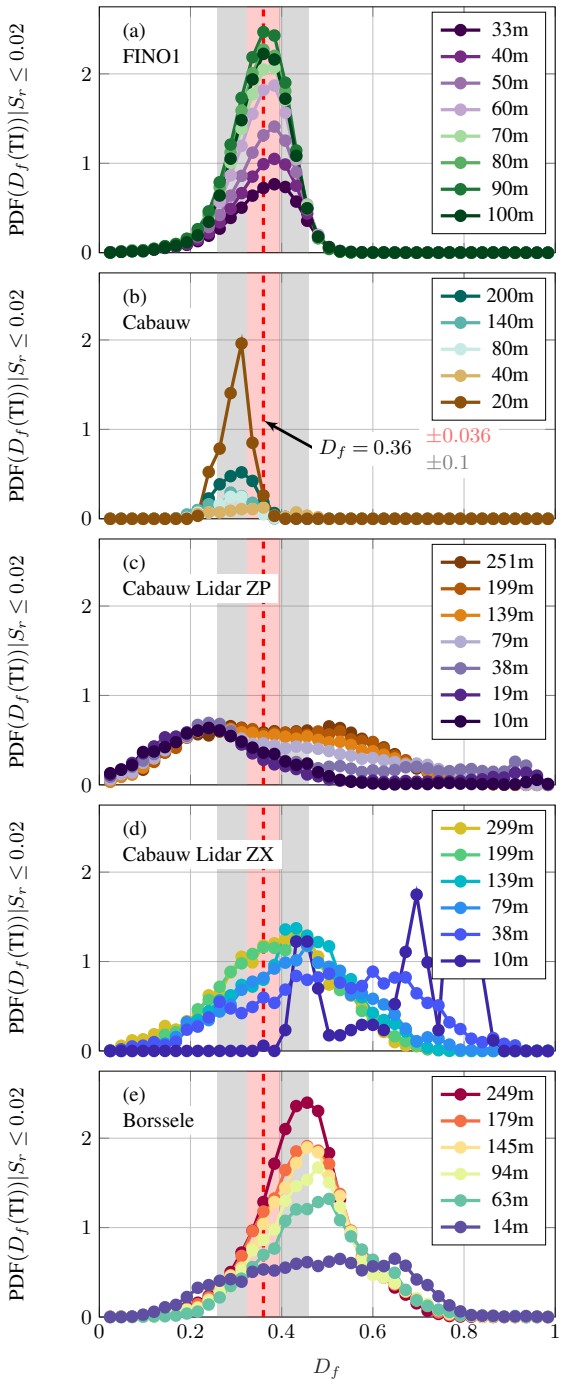

**Figure 12.** Probability density function (normalization according to Fig. 10) of the fractal dimension $D_f$ conditioned on the TI range $2.5\% < \text{TI} < 7.5\%$ for FINO1 (a), Cabauw (b), Cabauw Lidar ZP (c), Cabauw Lidar ZX (d), and Borssele (e). The red dashed line indicates the typical TNTI fractal dimension of 0.36 and the shaded red area a range of $\pm0.036$ (gray area $\pm0.1$) around this value. For a further quantification, see Fig. 13.

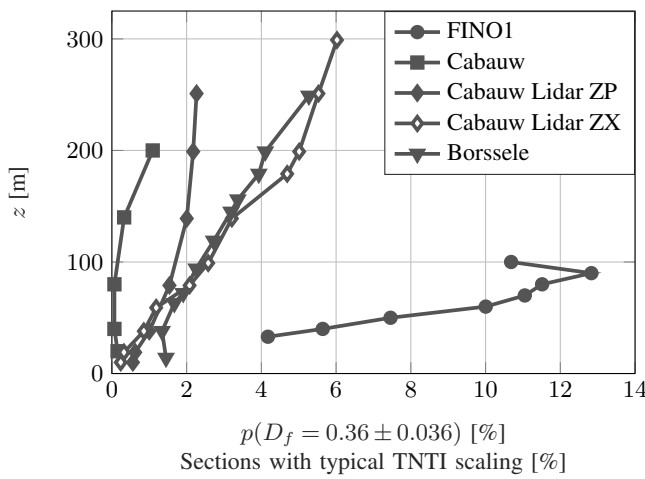

**Figure 13.** Percentage of data exhibiting a typical TNTI fractal dimension.

the investigated scales (two decades) are considered. If the fractality changes over the investigated scales, the fractal dimensions are neglected. Hence, if a partially typical TNTI fractal dimension would be considered, even higher amounts of sections with a typical TNTI fractal dimension might be found.

As a side remark, we would like to point out that an increased probability of fractal dimensions in the order of 2/3 is observed in the lidar measurements at low altitudes (see Fig. 12 (c-e)). This could be interpreted as a consequence of thermally driven (convective) flow fields exhibiting a 2/3 scaling (Grossmann and Lohse, 1994).

Differences are observed at different measurement locations and for different measurement techniques, including temporal resolution, spatial resolution, and observed periods. The resolution of the measurement is important to get proper values. As the fractality describes the self similarity on different scales, the temporal (or spatial) resolution defines the lower bound until which fractal features can be seen. While the met masts give information on the small scales (below the rotor diameter), the lidar data sets only give information on larger scales. For the investigated frequencies a robust behavior of the fractality is observed. In Fig. 11, reduced probabilities are observed at $100\,\mathrm{m}$, which do not follow the trends. This phenomenon, that the statistics of the measurement point at $100\,\mathrm{m}$ deviate from those at the other heights, is known but unexplained for the FINO1 data set.

At higher altitudes, more fractal subsets are seen. This is expected, as in the meteorological context the TNTI can be understood as the dynamic interface between the turbulent atmospheric boundary layer (commonly known as the Prandtl layer) and the laminar flow (which could be referred to as laminar Ekman layer) that occurs at higher altitudes. The estimated fractal dimension of the TNTI accumulates for all data sets around a certain value, which is in a first order approximation close to 0.36, the reference value of ideal laboratory experiments. Lidar measurements, which cover different (larger) scales, also show an accumulation of the fractal dimension at a certain value, suggesting a universal meaning of the fractality of the TNTI. However,

deviations ($\pm 0.1$) of the fractal dimension are found, which could be due to effects resulting from different orographies and measuring methods and need to be further investigated.

## 6   Conclusions

The presence of the turbulent/non-turbulent interface (TNTI) in the atmosphere at different sites has been studied. Our results
of fractal dimension of $0.36 \pm 0.1$ we take as strong hint for comparable trends for the different measurements sites.

The fractal dimension, a simple multi-scale approach, provides an effective method for characterizing the complexity of the TNTI. The typical fractal dimension of the TNTI of 0.36 known from laboratory experiments is quite close to the values found in the atmosphere. The highest likelihood for a typical TNTI fractal dimension is found at high altitudes. Hence, the geometry of the TNTI for atmospheric cases and more ideal flow situations in laboratory experiments and numerical simulations seems
to be quite similar. This opens up new possibilities for further detailed studies.

Independent of the measurement location and procedure, a significant amount of sections with a typical TNTI fractal dimension is detected. Our analysis of several data sets reveals that the fractality of the TNTI occurs at very different scales, from the size of a wind turbine blade to several kilometers (as seen in lidar data). Up to more than 10% of the observed time, a TNTI at small (for a wind turbine relevant) scales is present at heights above $60 \, \mathrm{m}$ (offshore, FINO1). This hints on a very frequent
presence of the TNTI at altitudes of a multi megawatt wind turbine rotor.

Further and more detailed investigations need to be made to get a complete picture of the TNTI in the atmosphere. High spatial and temporal resolved data over long periods are needed to gain further knowledge on its small scale behavior.

These findings make the consideration of laminar flows and the frequent presence of the TNTI at higher altitudes relevant for wind turbine research. This is particularly important for large offshore wind turbines in the multi megawatt class. The sudden
jump between two significantly different turbulence states could cause additional load cycles for the turbine components. Experimental and numerical studies are needed to investigate the effects of the TNTI on wind turbines and to clarify whether the TNTI needs to be considered in turbine design and operation. For this purpose, an indicated universal structure of the TNTI is very helpful.

*Data availability.*   Wind data for the Cabauw and Borssele site were made available by the Royal Netherlands Meteorological Institute (KNMI)

**Appendix A:  Filter span**

Changing the filter span $T$ can significantly influence the results. Since the fluctuations are determined by subtracting a moving average velocity from the velocity time series, a filter span $T$ that is too small would lead to a subtraction of relevant fluctuations and, in extreme cases, a purely laminar time series would remain. The chosen moving average window size of $20 \, \mathrm{s}$ comes from the largest (3D) turbulent structures found in the atmosphere, which are of the order of $0.05 \, \mathrm{Hz}$ (see Sim et al., 2023). This

frequency gives the largest turbulence length scale of $20\,\text{s}$. We have chosen a filter span $T$ that correspond to the large scale turbulence structures for the mean wind speed and thus "highpass filtered" our results on the largest relevant scales. This also makes our results comparable to wind tunnel studies where no wind speed fluctuations occur at such scales.

A systematic analysis on the influence of the filter span $T$ on the fractal dimension $D_f$ is shown in Fig. A1. The cases Fig. A1 (a-c) and Fig. A1 (d-f) correspond to the $60\,\text{m}$ cases shown in Fig. 6 (a) and Fig. 6 (b) (08.05.2008 4:00 and 16:00), respectively. Up to a $T$ of roughly $20\,\text{s}$ a variation of the number of boxes $N_{\text{Box}}$ and the estimated fractal dimension $D_f$ can be recognized. For larger $T$ only small deviations occur. This confirms our choice of the filter span $T$. A filter span $T < 20\,\text{s}$ that is too small filters out relevant fluctuations and affects the analysis of the fractal dimension, whereas the method becomes robust for larger scales.

The lidar data sets exhibit a lower sampling frequency, so a deviation from this scale was necessary. A compromise between a sufficient amount of samples for the estimation of the turbulent kinetic energy and sufficiently small scales was found for a window size of $90\,\text{s}$. This value is close to the kink between "wall turbulence" and "3D turbulence" defined by Sim et al. (2023) and is therefore still dominated by 3D turbulence. For the lidar data sets a similar behavior was found for $T > 90\,\text{s}$.

## Appendix B:  Section length

Investigations on 10 minute sections are a common approach in the field of wind energy. For our analysis, it was found that a sufficient amount of data is available for the analysis in a 10 minute section. For the appropriate length Sreenivasan and Meneveau (1986) found that the window sizes should be below 50 integral time scales to show fractal-like behavior, while on larger scales random behavior with a fractal dimension of 1 occurred. In our case, 50 integral time scales correspond to $1000\,\text{s}$, which is close to the $600\,\text{s}$ we chose. Therefore, we assume our section length to be appropriate.

Fig. B1 shows the influence of section length $T_{\text{sec}}$ on the analysis of the fractal dimension. The cases Fig. B1 (a-c) and Fig. B1 (d-f) correspond to the $60\,\text{m}$ cases shown in Fig. 6 (a) and Fig. 6 (b) (08.05.2008 4:00 and 16:00), respectively. As expected, the number of boxes $N_{\text{Box}}$ increases with the section length. However, the trend of the curves $N_{\text{Box}}(r_{\text{Box}}$ is hardly influenced and only differs for short section lengths ($T_{\text{sec}} < 600\,\text{s}$. For longer section lengths, the characteristics tend to converge to a certain value for the fractal dimension $D_f$ as well as for the residual standard error $S_r$. However, for longer section lengths more and different flow characteristics are considered and an average value is extracted. Hence, a section length of $600\,\text{s}$ seems to be a good compromise between a sensitive behavior on small section length changes and averaging over a long duration.

For the lidar measurements, longer sections ($5400\,\text{s}$ corresponding to roughly 300 integral time scales) were considered due to the lower temporal resolution. However, for these cases we shifted the upper spatial limit for determining the fractal dimension by a factor of 10. Hence, we are shifting the largest investigated scales and hence again having a section length which is in the order of 50 times of the largest investigated scales.

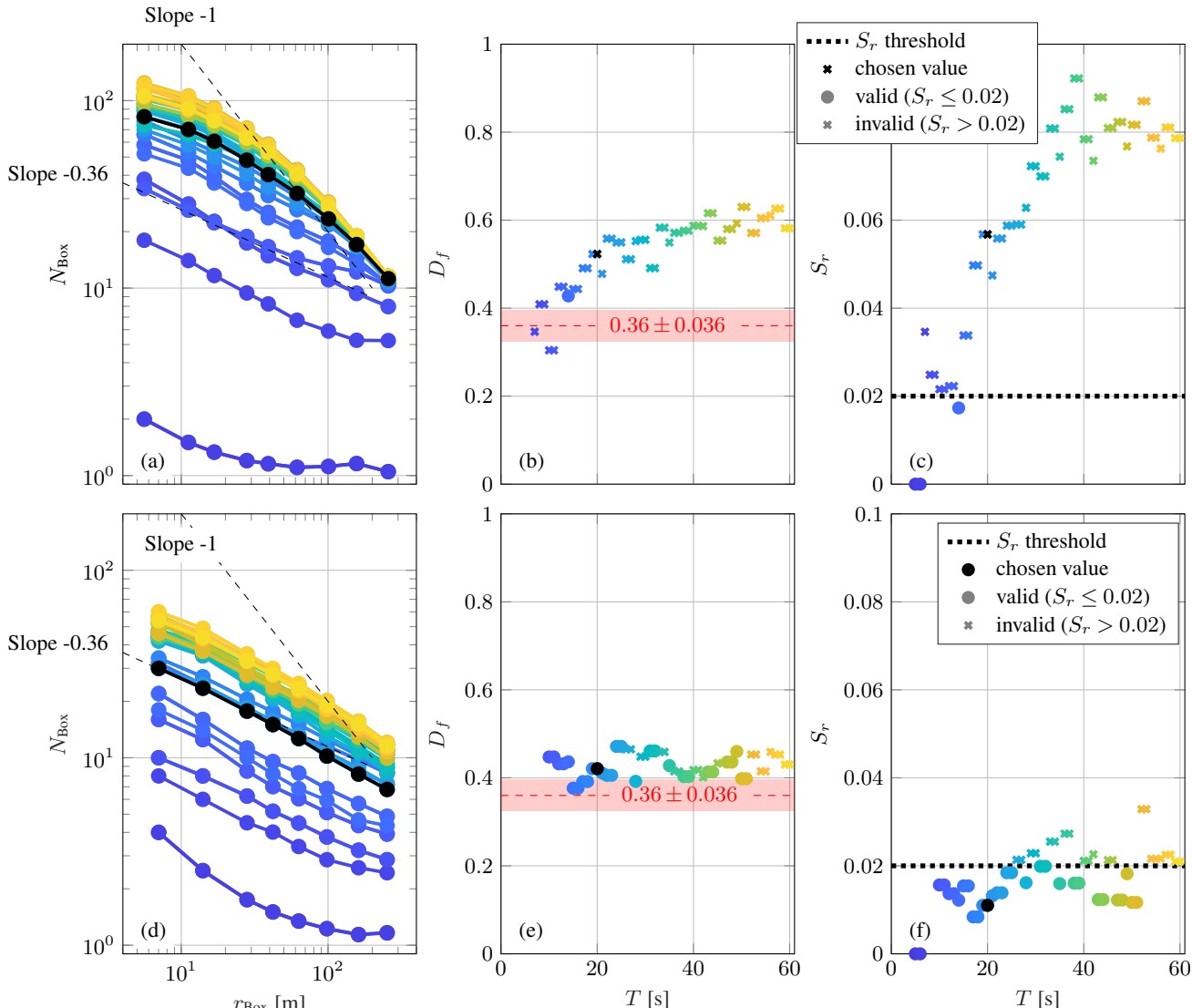

**Figure A1.** Influence of the filter span $T$ on the number of boxes $N_{\text{Box}}$(a,d), the fractal dimension $D_f$ (b,e), and the residual standard error $S_r$ (c,f), respectively for a rather turbulent section (a-c) and a section with TNTI characteristics (d-f). The Colors (a, d) indicate the filter span $T$ from blue for low to yellow for high values as shown in (b), (c), (e), and (f). In black the results for the chosen filter span $T$ of $20\,\text{s}$ are shown.

## Appendix C: Intermittency factor

In Fig. C1 the intermittency factor

$$\gamma = \frac{4}{F} \tag{C1}$$

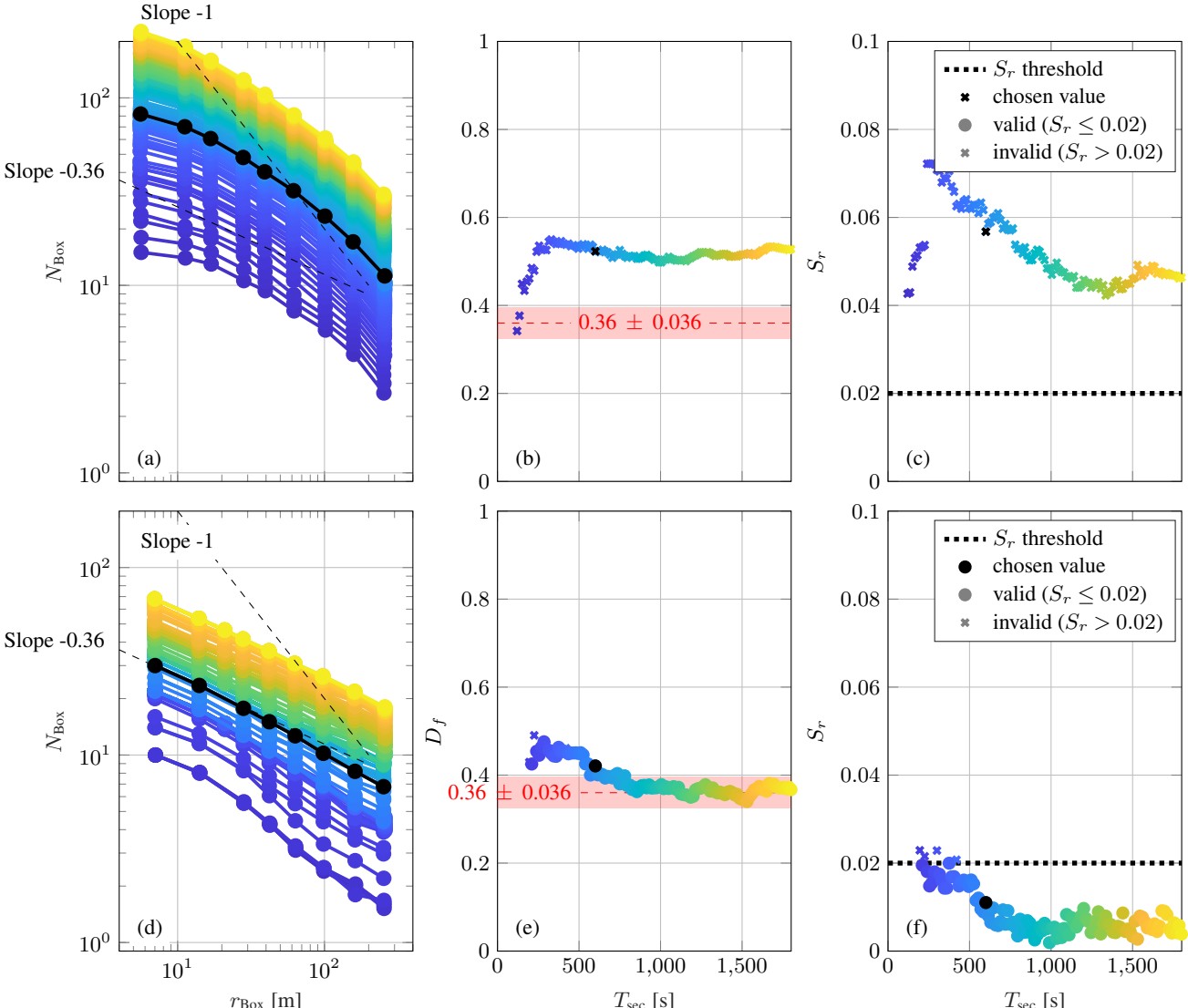

**Figure B1.** Influence of the section length $T_{\text{sec}}$ on the number of boxes $N_{\text{Box}}$(a,d), the fractal dimension $D_f$ (b,e), and the residual standard error $S_r$ (c,f), respectively for a rather turbulent section (a-c) and a section with TNTI characteristics (d-f). The Colors (a, d) indicate the section length $T_{\text{Sec}}$ from blue for low to yellow for high values as shown in (b), (c), (e), and (f). In black the results for the chosen section length $T_{\text{sec}}$ of 600 s are shown.

after Townsend (1951) with the flatness $F = \langle u_\tau^4 \rangle / \langle u_\tau^2 \rangle^2$ of the velocity increments $u_\tau = u(t) - u(t + \tau)$ for the smallest

possible time interval $\tau = 1/f_s$ defined by the sampling frequency $f_s$ for the different data sets in function of the height $z$ is

shown. A value of $\gamma = 1$ indicates turbulent flow, whereas 0 denotes laminar flow. The data sets show comparable trends with

a decrease of $\gamma$ with height. The Cabauw data set deviates from this trend and exhibits very low $\gamma$ trough out. The FINO1 data

set shows an outlier at 80m, which is not further analyzed here ($100\,\mathrm{m}$ deviates from the FINO1 trend as discussed in Sect. 5). The intermittency factor at low heights already exhibits comparatively low values, which may be caused by the presence of laminar phases at lower heights.

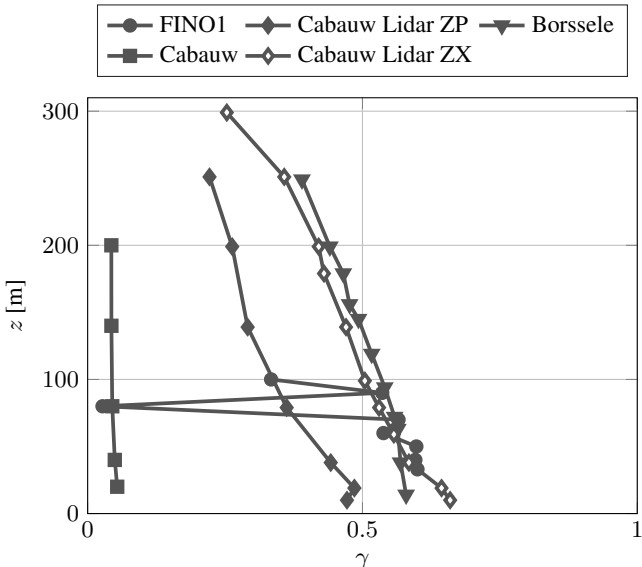

**Figure C1.** Intermittency factor $\gamma$ dependent on height $z$.

*Author contributions.* LN analyzed the data and wrote the manuscript draft. MW and JP supervised the work and reviewed and edited the manuscript.

*Competing interests.* At least one of the (co-)authors is a member of the editorial board of *Wind Energy Science*.

*Acknowledgements.* We acknowledge helpful discussions with Michael Hölling, Fabien Thiesset and Jan Friedrich. The project has been
funded by the Deutsche Forschungsgemeinsschaft (DFG, German Research Foundation) – SFB1463 – 434502799.

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
