# Peer review of "The fractal turbulent/non-turbulent interface in the atmosphere"

_Wind Energy Science, 2023_

## Referee Comment (RC1)

Report on paper "The fractal turbulent/non-turbulent interface in the atmosphere"
by L. Neuhaus, M. Wächter, and J. Peinke

The authors present a detailed analysis of the geometrical, possibly fractal, properties of the atmospheric turbulent/non-turbulent interface (TNTI) by analyzing data from wind speed measurements at three different locations (FINO1, Cabauw, Borssele Alpha). I found the paper instructive, well written, and I appreciated the clarity of the graphs despite the large amount of information they provide. Overall, this study provides scientifically sound results and conclusions.

Here are few suggestions that aim at improving the clarity of the presentation and providing some simple, yet important, results about the TNTI. All the points below are given in chronologic order. Points 4 & 5 below is to me the most critical suggestions I would like the authors to address.

1. Section 2, it was not clear to me if the anemometers data issuing from either the FINO1 or the Cabauw were obtained simultaneously at different heights. If yes, that means that 2 dimensional maps of the wind speed as a function of time, $t$, and height, $z$, could have been constructed and used to tackle a 2D analysis of the fractal properties of the TNTI. Am I right? I agree that the resolution in z-direction is probably not sufficient for such an analysis to be carried out but maybe this could be written somewhere in section 2.

2. Section 3.1:
   a. In section 3.1, the authors present the state-of-the-art of TNTIs. When discussing the work by Sreenivasan and Meneveau (1986), the authors could add that Sreenivasan and Meneveau discovered that a fractal scaling can exist in an intermediate range of scales which is comprised between an inner cutoff (a small scale which they found to be of order of the Kolmogorov scale) and an outer cutoff (a large scale which is generally assumed to be proportional to the integral length-scale).
   b. the phrase at line 72 starting with "For a reduction…" could be deleted since this will be more clearly explained at the level of section 3.2, Eq. (2).
   c. I feel that it could be worth recalling that that a surface has a fractal dimension which is bounded, i.e. $2<D\_f<3$. A surface with dimension=2 is smooth (e.g. a sphere has a surface which grows with power 2 of its diameter) while a surface with a fractal dimension of 3 is so tortuous that it fills the entire space. For scales below the inner-cutoff (see point (a)), viscosity tends to smooth out the interface and the surface becomes smooth (with dimension 2).

3. Section 3.2, to be more precise, it should be mentioned that the dimension which is measured using the box counting method is the "box dimension" or Minkowski-Bouligand dimension. This may differ with other measures of fractality using say the caliper technique, spectra or correlation functions.

4. Section 3.3, the authors have used a time window of 20s (90s for the Lidar measurements) for computing the moving average velocity and related TI. Could the

authors justify this choice? Are results sensitive to this parameter? Similarly, the authors analyze statistical results for the TNTI based on a 10 minutes window. Could you please justify this choice and provide material and discussions on how results change when this window is increased/decreased?

5. In section 4, I regret that the analysis the authors have performed is not able to answer the straightforward question of the height at which the TNTI is located. For doing this, the authors could have showed the number of crossings per unit time (or unit length given the Taylor hypothesis) as a function of height. This represents the probability of finding the TNTI at a given location. In 3D, this is the surface density. My opinion is that it is the first quantity that should be presented and discussed in section 4. In the context of wind energy production, I think it gives a good idea of the relative position between the atmospheric TNTI and the height of the wind turbine. Similarly, the authors could provide the portion of time the signal is in turbulent state versus laminar state as a function of $z$. In the fluid mechanics community, this metric is sometimes referred to as the intermittency coefficient as defined by e.g Townsend. Here again, my feeling is that this is worth being documented in the context of wind energy production.

6. In Figs. 10, & 12, it does not seem that the pdfs are normalized in such a way the integral is one. Am I right? Should not they be normalized?

Typos:
- Line 148, "superposition"
- Please rephrase "the results seem to get physical unreasonable"
- Line 184, maybe it should be added that the shear exponent will be defined later.
- Line 226, the word "astonishing" is an appreciation of the authors about their own work, "in good agreement" alone is more neutral.
- Conclusions, line 255, I think some words have been swept. Please reword.

---

## Author Comment (AC1)

Dear Anonymous Referee,

thank you for reviewing our paper draft, your positive feedback and all the thoughtful and helpful comments.

In the following we try to answer your questions and consider your remarks. Comments of the referee are in **bold font**, replies are given in regular font and our adaptions in the letter are shown. Proposed changes in the text of the paper are marked by *italic font*. Added content is highlighted by *blue and underlined* and deleted content by *red and crossed out*.

Yours sincerely,

Lars Neuhaus (on behalf of all authors)
* * *
**The authors present a detailed analysis of the geometrical, possibly fractal, properties of the atmospheric turbulent/non-turbulent interface (TNTI) by analyzing data from wind speed measurements at three different locations (FINO1, Cabauw, Borssele Alpha). I found the paper instructive, well written, and I appreciated the clarity of the graphs despite the large amount of information they provide. Overall, this study provides scientifically sound results and conclusions. Here are few suggestions that aim at improving the clarity of the presentation and providing some simple, yet important, results about the TNTI. All the points below are given in chronologic order. Points 4 & 5 below is to me the most critical suggestions I would like the authors to address.**

Thank you for your positive feedback. We will answer your questions in the chronologic order.
* * *
**1. Section 2, it was not clear to me if the anemometers data issuing from either the FINO1 or the Cabauw were obtained simultaneously at different heights.**

The data was recorded simultaneously, this we now state in the revised version.

Page 2, lines 37 – 38:

*[...] Cup anemometer at* $33\,\mathrm{m}$, $40\,\mathrm{m}$, $50\,\mathrm{m}$, $60\,\mathrm{m}$, $70\,\mathrm{m}$, $80\,\mathrm{m}$, $90\,\mathrm{m}$, *and* $100\,\mathrm{m}$ *record the wind speed simultaneously with a sampling frequency of 1Hz. [...]*

Page 2, lines 46 – 47:
*[...] Propeller anemometer at* $20\,\mathrm{m}$, $40\,\mathrm{m}$, $80\,\mathrm{m}$, $140\,\mathrm{m}$, *and* $200\,\mathrm{m}$ *record the wind speed simultaneously with a sampling frequency of 2Hz. [...]*

**If yes, that means that 2 dimensional maps of the wind speed as a function of time, t, and height, z, could have been constructed and used to tackle a 2D analysis of the fractal properties of the TNTI. Am I right? I agree that the resolution in z-direction is probably not sufficient for such an analysis to be carried out but maybe this could be written somewhere in section 2.**

Yes, the comparatively small height region and the height resolution are too low for a meaningful box-counting analysis. Hence, the 1D approach is favored here. We now mention this in the manuscript.

Page 4, lines 97 – 100:
*In real-world applications, [...]  Atmospheric data in particular is mostly only available by point wise measurements. The limited amount of vertical measurement points of the investigated data sets is not sufficient for a two-dimensional analysis. However, by Taylor's hypothesis of frozen turbulence (Taylor, 1938)  the individual point measurements will give a one-dimensional slice through a three-dimensional field. [...]*
* * *
**2. Section 3.1: a. In section 3.1, the authors present the state-of-the-art of TNTIs. When discussing the work by Sreenivasan and Meneveau (1986), the authors could add that Sreenivasan and Meneveau discovered that a fractal scaling can exist in an intermediate range of scales which is comprised between an inner cutoff (a small scale which they found to be of order of the Kolmogorov scale) and an outer cutoff (a large scale which is generally assumed to be proportional to the integral length-scale).**

Thank you for this comment, we highlight this finding now.

Page 3, lines 71 – 73:
*[...] By changing the  image resolution, different scales were resolved and *

* on an intermediate range of scales between the Kolmogorov length scale and 1/6 integral length scale L a fractal dimension of the TNTI of about 2.4 was found.*
* * *
**2. Section 3.1: b. the phrase at line 72 starting with "For a reduction..." could be deleted since this will be more clearly explained at the level of section 3.2, Eq. (2).**

We deleted the corresponding sentence.

Page 3, line 73:
**
* * *
**2. Section 3.1: c. I feel that it could be worth recalling that that a surface has a fractal dimension which is bounded, i.e. $2 < D_f < 3$. A surface with dimension=2 is smooth (e.g. a sphere has a surface which grows with power 2 of its diameter) while a surface with a fractal dimension of 3 is so tortuous that it fills the entire space. For scales below the inner-cutoff (see point (a)), viscosity tends to smooth out the interface and the surface becomes smooth (with dimension 2).**

Thank you for this comment. This will help to further understand the embedding dimension. We added a comment on the boundaries of the fractal dimension.

Pages 4 – 5, lines 100 – 106:
*[...]  By the additive rule of co-dimensions for intersecting sets*

$$D_{f,3} = D_{f,2} + 1 = D_{f,1} + 2 \tag{1}$$

*the fractal dimension $D_{f,d}$ in higher embedding dimensions $d$ can be estimated by data collected in a lower embedding dimensions (see Mandelbrot, 1982; Sreenivasan and Meneveau, 1986). Furthermore, the fractal dimension is bounded by the embedding dimension $d$ and the corresponding lower dimension $d-1$, e.g. a smooth surface in three-dimensional space would scale with $D_f = 2$, whereas a space-filling surface would exhibit a fractal dimension of $D_f = 3$.*

**3. Section 3.2, to be more precise, it should be mentioned that the dimension which is measured using the box counting method is the "box dimension" or Minkowski-Bouligand dimension. This may differ with other measures of fractality using say the caliper technique, spectra or correlation functions.**

We added a corresponding comment.

Page 4, lines 91 – 96:
*The fractality  of this Koch curve can be estimated by a box counting approach, which results in the fractal dimension (box-counting dimension or Minkowski–Bouligand dimension). To do so, boxes with different edge length $r$ are used and the  number of boxes $N(r)$  required to cover the curve  is counted. The fractal dimension $D_f$  (box-counting dimension) can then be determined by the slope of the relation*

$$N(r) \propto r^{-D_f} \tag{2}$$

*to 1.262 for the Koch curve (see Sreenivasan and Meneveau, 1986).*
* * *
**4. Section 3.3, the authors have used a time window of 20s (90s for the Lidar measurements) for computing the moving average velocity and related TI. Could the authors justify this choice? Are results sensitive to this parameter? Similarly, the authors analyze statistical results for the TNTI based on a 10 minutes window. Could you please justify this choice and provide material and discussions on how results change when this window is increased/decreased?**

This are indeed good questions and we did not mention our thoughts here. We added some analysis on the effect of the filter span and the section length in the appendix (Appx. A, page 18 and Appx. B, page 19) and added comments in the manuscript.

Pages 5 – 6, lines 122 – 131:
*The instantaneous TKE is approximated by*

$$E = \frac{1}{2}(u - u_{movavg})^2 \tag{3}$$

*with the moving averaged wind speed*

$$u_{movavg} = \frac{1}{Tf_s} \sum_{\Delta t=-T/2}^{T/2} u(t + \Delta t).$$ (4)

*Here the sampling frequency $f_s$ and the filter span $T$ of $20\,\text{s}$ (for cup and propeller anemometer) and  $90\,\text{s}$ (for Lidar measurements) are used. These values are chosen, as they mark the boundary between 3D turbulence and large scale fluctuations (see Sim et al., 2023). For the lidar measurements a larger window size is considered as a compromise between a sufficient amount of samples for the estimation of the TKE and sufficiently small scales. To validate that choice, we performed a study on the influence of a variation of $T$, which showed no significant changes for $T > 20\,\text{s}$ and thus shows a robust behavior for changes on large scales (see Appx. A).*

Page 6, lines 137 – 139:
*The next steps are shown examplarily for a day (May 8, 2008) of the FINO1 data set (Fig. 4), as this day exhibits many laminar periods. The investigation is done for sections of 10 minute length (sensitivity on section length is shown in Appx. B). In Fig. 5 crossings of the TNTI are visualized for the different heights.*
* * *
**In section 4, I regret that the analysis the authors have performed is not able to answer the straightforward question of the height at which the TNTI is located. For doing this, the authors could have showed the number of crossings per unit time (or unit length given the Taylor hypothesis) as a function of height. This represents the probability of finding the TNTI at a given location. In 3D, this is the surface density. My opinion is that it is the first quantity that should be presented and discussed in section 4. In the context of wind energy production, I think it gives a good idea of the relative position between the atmospheric TNTI and the height of the wind turbine.**

That is correct. We do not intend to determine the height of a boundary layer structure. Our analysis only provides information on whether the TNTI reaches the height measured or not. From the statistics it can be seen how often this happens. This is an important point that we have now clarified in the manuscript.

Page 15, lines 236 – 238:

*A frequent presence of the turbulent/non-turbulent interface (TNTI) in the atmospheric data is observed. The presented method provides information on how frequently TNTI features occur at the investigated heights, but does not allow the height position of the TNTI to be determined. [...]*

**Similarly, the authors could provide the portion of time the signal is in turbulent state versus laminar state as a function of z. In the fluid mechanics community, this metric is sometimes referred to as the intermittency coefficient as defined by e.g Townsend. Here again, my feeling is that this is worth being documented in the context of wind energy production.**

This is a good idea. We now included Fig. 9(b) in the manuscript showing the amount of sections with a TI < 1.5%. Furthermore, we added an analysis of the intermittency coefficient in the appendix (Appx. C, page 20).

Page 10, lines 174 – 176:

*The  general trend towards a lower TI at higher  altitudes is illustrated by a decrease of the median  TI (med(TI)) and an increase in the portion of 10 minute sections with TI$< 1.5\%$ as a function of z ( Fig. 9, see Appx. C for an analysis of the intermittency factor $\gamma$). [...]*
* * *
**6. In Figs. 10, & 12, it does not seem that the pdfs are normalized in such a way the integral is one. Am I right? Should not they be normalized?**

Yes. We mentioned the normalization in the caption and clarified this in the text now.

Page 10, lines 182 – 184:

*Next the fractal dimension of the TNTI is investigated for 10 minute sections with an overlap of 9 min. Figure 10 shows the individual probability density function (PDF) of the fractal dimension $D_f$ for different TI ranges. The PDFs are normalized including invalid fractal dimensions ($S_r > 0.02$), which are not shown but would correspond to a peak in the PDF at "NaN".*

Page 15, lines 215 – 217:

*Figure 12 shows the distribution of the fractal dimension for *

*each data set, according to Fig. 10 (b) for 2.5% < TI < 7.5%. The PDFs are normalized including invalid fractal dimensions ($S_r > 0.02$), which are not shown but would correspond to a peak in the PDF at "NaN".[...]*
* * *
**Typos**

Thank you for noticing. We corrected the corresponding Typos.
* * *
Thank you very much for your efforts and your thoughtful comments,

Lars Neuhaus (on behalf of all authors)

---

## Author Comment (AC2)

Dear Anonymous Referee,

thank you for reviewing our paper draft, your positive feedback and all the thoughtful and helpful comments.

In the following we try to answer your questions and consider your remarks. Comments of the referee are in **bold font**, replies are given in regular font and our adaptions in the letter are shown. Proposed changes in the text of the paper are marked by *italic font*. Added content is highlighted by *blue and underlined* and deleted content by .

Yours sincerely,

Lars Neuhaus (on behalf of all authors)
* * *
**This paper presents a study to investigate the Turbulent/Nonturbulent interface in the atmosphere using measurements from met masts and lidars on two offshore and one onshore windsite. Existence of the TNTI interface and its probability distribution with height as well as its fractal scaling is studied. The paper is well written in general.**

Thank you for your positive feedback. We will answer your questions in the chronologic order.

**Use of English language needs improvement as to some expressions and descriptions sound strange. Some comments are below:**

We are sorry, that the first version was not as good as we had hoped. We have made an effort to identify poorly worded expressions and improve the language. All changes can be found in the 'diff file'.
* * *
**In Section 4.1, authors mention that strong influence of measurements techniques are expected but they indicate that this is out of scope for this study. I understand that but I think the authors should elaborate somewhat more since these differences may influence the results presented in this paper. The anemometers sample at 1 Hz and 2 Hz sampling rates but the Lidars provide temporal data with a resolution at every 17 or 18 s. So there is a big differ-**

**ence in sampling rates. Please provide some comments regarding the effects of these big differences on presented results.**

Thank you for your hint. We removed the misleading comment and now explain the effect of the different sampling frequencies.

Page 10, lines 172 – 173:
Figure 8 shows the resulting probability density functions (PDF) for the individual sites. All sites show an increase of low turbulence intensity  sections with height.

Page 17, lines 252 – 257:
*Differences are observed at different measurement locations and for different measurement techniques, including temporal resolution, spatial resolution, and observed periods. The resolution of the measurement is important to get proper values. As the fractality describes the self similarity on different scales, the temporal (or spatial) resolution defines the lower bound until which fractal features can be seen. While the met masts give information on the small scales (below the rotor diameter), the lidar data sets only give information on larger scales. For the investigated frequencies a robust behavior of the fractality is observed. […]*
* * *
**In Figure 8c, the data shows very wide pdf distributions at low heights unlike other Cabauw data. Authors allude that this could be due to differences in measurement methods. Please elaborate. Why are there significant characteristic differences in pdf distributions between the metmast data and the Lidar data at Cabauw?**

All data is from different measurement campaigns (see Sec. 2 of the manuscript) and based most likely on different meteorological conditions. We are now clarifying this in the manuscript.

Page 10, lines 178 – 180:
*[…] However, a direct comparison is difficult due to the different measurement methods, the different measurement periods and seasons. Thus these statistics are based on different meteorological conditions which were selected..*

**In Figure 8a the color scale is poorly chosen. It's hard to distinguish 100 m and 33 m data for example.**

We are sorry for this inconvenience. The problem we have is that we chose a color scheme that we used for all results to show them consistently. Since we intend to show with this figure a common PDF for all heights, we think the message is still delivered despite the problem of seeing the difference between 100m and 33m. In addition, quantified details are given in Fig. 9. We now mention this in the caption of Fig. 8. To remain consistent, we would like to leave this figure unchanged.

If the referee is not satisfied with our response, we offer to include individual figures per height in the appendix.

Page 11:
**Figure 8.** *Probability density functions (PDF) of the turbulence intensity at different heights for the data sets FINO1 (a), Cabauw (b), Cabauw Lidar ZP (c), Cabauw Lidar ZX (d), and Borssele (e). For a further quantification, see Fig. 9.*

Page 13:
**Figure 10.** *Probability density function of the fractal dimension $D_f$ conditioned on the different TI ranges: $TI < 2.5\%$ (a), $2.5\% < TI < 7.5\%$ (b), and $TI > 7.5\%$ (c). The red dashed line indicates the typical TNTI fractal dimension of 0.36 and the shaded red area a range of $\pm 0.036$ around this value. The normalization of the PDFs is done based on all sections including invalid fractal dimensions ($S_r > 0.02$), which are not shown but would correspond to a peak at "NaN". For a further quantification, see Fig. 11 (a).*

Page 16:
**Figure 12.** *Probability density function (normalization according to Fig. 10) of the fractal dimension $D_f$ conditioned on the TI range $2.5\% < TI < 7.5\%$ for FINO1 (a), Cabauw (b), Cabauw Lidar ZP (c), Cabauw Lidar ZX (d), and Borssele (e). The red dashed line indicates the typical TNTI fractal dimension of 0.36 and the shaded red area a range of $\pm 0.036$ (gray area $\pm 0.1$) around this value. For a further quantification, see Fig. 13.*

Thank you very much for your efforts and your thoughtful comments,

Lars Neuhaus (on behalf of all authors)